# DEMYSTIFYING GRAPH NEURAL NETWORKS VIA GRAPH FILTER ASSESSMENT

## ABSTRACT

Graph Neural Networks (GNNs) have recently received tremendous attention due to their power in handling graph data for different downstream tasks across different application domains. Many GNN models have been proposed, which mainly differ in their graph filter design. However, most of these models believe there is a best filter for all the graph data. Instead, we attempt to provide in-depth analysis on *(1) Whether there exists an optimal filter that performs the best on all graph data; (2) Which graph properties should be considered for finding the optimal graph filter; and (3) How to design appropriate filters that adapt to a given graph.* In this paper, we focus on addressing the above three questions for the semi-supervised node classification task. We propose a novel assessment tool, called Graph Filter Discriminant Score (GFD), for evaluating the effectiveness of graph convolutional filters for a given graph in terms of node classification. Using the assessment tool, we find out that there is no single filter as a "silver bullet" that performs the best on all possible graphs, and graphs with different properties are in favor of different graph convolutional filters. Based on these findings, we develop Adaptive Filter Graph Neural Network (AFGNN), a simple but powerful model that can adaptively learn data-specific filters. For a given graph, AFGNN leverages graph filter assessment as an extra loss term and learns to combine a set of base filters. Experiments on both synthetic and real-world benchmark datasets have demonstrated that our proposed model has the flexibility in learning an appropriate filter and consistently provides state-of-the-art performance across all the datasets.

## 1 INTRODUCTION

Graph Neural Networks (GNNs) are a family of powerful tools for representation learning on graph data, which has been drawing more and more attention over the past several years. GNNs can obtain informative node representations for a graph of arbitrary size and attributes, and has shown great effectiveness in graph-related down-stream applications, such as node classification (Kipf & Welling, 2017), graph classification (Wu et al., 2019b), graph matching (Bai et al., 2019), recommendation systems (Ying et al., 2018), and knowledge graphs (Schlichtkrull et al., 2018).

As GNNs have superior performance in graph-related tasks, the question as to what makes GNNs so powerful is naturally raised. Note that GNNs adopt the concept of the convolution operation into graph domain. To obtain a representation of a specific node in a GNN, the node aggregates representations of its neighbors with a convolutional filter. For a task related to graph topology, the convolutional filter can help GNN nodes to get better task-specific representations (Xu et al., 2019). Therefore, it is the filter that makes GNNs powerful, and thus the key to designing robust and accurate GNNs is to design proper graph convolutional filters.

Recently, many GNN architectures are proposed (Zhou et al., 2018) with their own graph filter designs. However, none of them have properly answered the following fundamental questions of GNNs: *(1) Is there a best filter that works for all graphs? (2) If not, what are the properties of graph structure that will influence the performance of graph convolutional filters? (3) Can we design an algorithm to adaptively find the appropriate filter for a given graph?*

In this paper, we focus on addressing the above three questions for semi-supervised node classification task. Inspired by studies in Linear Discriminant Analysis (LDA), we propose a Graph Filter

Discriminant (GFD) Score metric to measure the power of a graph convolutional filter in discriminating node representations of different classes on a specific graph. We have analyzed all the existing GNNs' filters with this assessment method to answer the three aforementioned questions. We found that no single filter design can achieve optimal results on all possible graphs. In other words, for different graph data, we should adopt different graph convolutional filters to achieve optimal performance. We then experimentally and theoretically analyze how graph structure properties influence the optimal choice of graph convolutional filters.

Based on all of our findings, we propose the Adaptive Filter Graph Neural Network (AF-GNN), which can adaptively learn a proper model for the given graph. We use the Graph Filter Discriminant Score (GFD) as a an extra loss term to guide the network to learn a good data-specific filter, which is a linear combination of a set of base filters. We show that the proposed Adaptive Filter can better capture graph topology and separate features on both real-world datasets and synthetic datasets.

We highlight our main contributions as follows:

- We propose an assessment tool: Graph Filter Discriminant Score, to analyze the effectiveness of graph convolutional filters. Using this tool, we find that no best filter can work for all graphs, the optimal choice of a graph convolutional filter depends on the graph data.
- We propose Adaptive Filter Graph Neural Network that can adaptively learn a proper filter for a specific graph using the GFD Score as guidance.
- We show that the proposed model can find better filters and achieve better performance compared to existing GNNs, on both real-word and newly created benchmark datasets.

## 2 PRELIMINARIES

**Semi-Supervised Node Classification.** Let $Y$ be the class assignment vector for all the nodes in $\mathbb{V}$. $C$ indicates the total number of classes, and $Y_v \in \{1, \cdots, C\}$ indicates the class that node $v$ belongs to. The goal of semi-supervised node classification is to learn a mapping function $f : \mathbb{V} \to \{1, \cdots, C\}$ using the labeled nodes, and predict the class labels for the unlabeled nodes, i.e., $\hat{Y}_v = f(v)$, by leveraging both node features $X$ and graph structure $A$.

**Graph Data Generator.** Intuitively, semi-supervised node classification requires both node features ($X$) and the graph structure ($A$) to be correlated to the intrinsic node labels ($Y$) to some extent. To systematically analyze the performance of different GNN filters, we test their performance under different graph data with different properties, i.e., graphs with different $X$, $A$, $Y$. Intuitively, both graph topology and node features have to be correlated with the node labels, if including both can enhance the performance of node classification task. To better understand the roles played by each component, we assume the graphical model to generate a graph data is as described in Fig. 1(a). To better disclose the relationship between different graph filters and properties of different graph data, we further make assumptions on how X and A are generated when Y is given, as it is difficult to obtain those properties from real-world data. Therefore, we study simulated data to support a thorough analysis. We now describe the generation of $Y$, $X|Y$, and $A|Y$ respectively.

**Generating $Y$:** Each node is randomly assigned with a class label with probability proportional to its class size. We assume each class $c$ is associated with $n_c$ nodes.

**Generating $X|Y$:** We assume that node features are sampled from a distribution determined by their corresponding labels. For example, we can sample node features of class $c$ from a multivariate Gaussian distribution with the parameters conditioned on class $c$: $X^{(c)} \sim \mathcal{N}(\boldsymbol{\mu}^{(c)}, \boldsymbol{\Sigma}^{(c)})$. For another example, we can sample node features of class $c$ from a circular distribution with radius $r_c$ and noise $noise_c$ conditioned on c.

**Generating $A|Y$:** We follow the most classic class-aware graph generator, i.e. stochastic block model (SBM, Holland et al. (1983)), to generate graph structure conditioned on class labels. SBM has several simple assumptions that (1) edges are generated via Bernoulli distributions independently and (2) the parameter of the Bernoulli distribution is determined by the classes of the corresponding pair of nodes $v_i$ and $v_j$, i.e., $A_{ij}|Y_i, Y_j \sim Ber(p_{Y_i Y_j})$, where $p_{Y_i Y_j}$ is a parameter determined by the two corresponding classes. In a simple two-class case, $p = p_{11} = p_{22}$ denotes the probability that the linked pair belongs to the same class, while $q = p_{12} = p_{21}$ denotes the probability that

the linked pair belongs to different classes. We call $\frac{p+q}{2}$ the *"density of graph"*, which controls the overall connectivity of a graph, and we call $|p - q|$ the *"density gap"*, which controls how closely the graph generated by SBM correlates with labels. We assume $p \geq q$ in all the following sections. Degree Corrected SBM (DCSBM, Karrer & Newman (2011)), which is a variation of SBM, adds another parameter $\gamma$ to control the *"power-law coefficient"* of degree distribution among nodes. Figure 1(b-e) demonstrates examples of synthetic graphs generated by SBM and DCSBM with different graph structure properties.

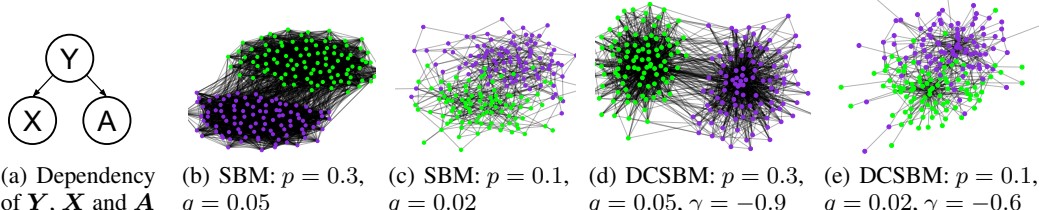

(a) Dependency of $\boldsymbol{Y}$, $\boldsymbol{X}$ and $\boldsymbol{A}$

(b) SBM: $p = 0.3$, $q = 0.05$

(c) SBM: $p = 0.1$, $q = 0.02$

(d) DCSBM: $p = 0.3$, $q = 0.05, \gamma = -0.9$

(e) DCSBM: $p = 0.1$, $q = 0.02, \gamma = -0.6$

Figure 1: (a) shows dependency between $\boldsymbol{Y}$, $\boldsymbol{X}$ and $\boldsymbol{A}$. (b) and (c) are dense and sparse graph generated by SBM, which have uniform degree distribution. (d) and (e) are dense and sparse graph generated by DCSBM, which have power-law degree distribution.

**Graph Convolutional Filters.** By examining various GNN designs, we find that most of the GNN operators can fit into a unified framework, i.e., for the $l$-th layer:

$$\boldsymbol{H}^{(l)} = \sigma(\mathcal{F}(\mathcal{G})\boldsymbol{H}^{(l-1)}\boldsymbol{W}) \tag{1}$$

which describes the three-step process that involves: (1) a graph convolutional operation (can also be regarded as feature propagation or feature smoothing) denoted as $\mathcal{F}(\mathcal{G})\boldsymbol{H}^{(l-1)}$, (2) a linear transformation denoted by multiplying $\boldsymbol{W}$, and (3) a non-linear transformation denoted by $\sigma(\cdot)$. Clearly, the graph convolutional operation $\mathcal{F}(\mathcal{G})\boldsymbol{H}^{(l-1)}$ is the key step that helps GNNs to improve performance. Thus, to design a good GNN, a powerful graph convolutional filter $\mathcal{F}(\mathcal{G})$ is crucial. We analyze the effectiveness of graph filters for existing GNNs in the following.

The work of GCN (Kipf & Welling, 2017) first adopts the convolutional operation on graphs and use the filter $\mathcal{F}(\mathcal{G}) = \tilde{\boldsymbol{D}}^{-1/2}\tilde{\boldsymbol{A}}\tilde{\boldsymbol{D}}^{-1/2}$. Here, $\tilde{\boldsymbol{A}} = \boldsymbol{A} + \boldsymbol{I}$ is the self-augmented adjacency matrix, and $\tilde{\boldsymbol{D}} = \text{diag}(\tilde{\boldsymbol{d}}_1, ..., \tilde{\boldsymbol{d}}_n)$ is the corresponding degree matrix, where $\tilde{\boldsymbol{d}}_i = \sum_{j=1}^{n} \tilde{\boldsymbol{A}}_{ij}$.

Some studies (Wu et al., 2019a; Maehara, 2019) use a filter $\mathcal{F}(\mathcal{G}) = (\tilde{\boldsymbol{D}}^{-1/2}\tilde{\boldsymbol{A}}\tilde{\boldsymbol{D}}^{-1/2})^k$ that is similar in form to GCN's filter, but with a pre-defined exponent $k$ greater than one. This would help a node to obtain information from its further neighbors without redundant computation cost. Several other studies propose to use sampling to speed up GNN training (Chen et al., 2018b; Hamilton et al., 2017; Chen et al., 2018a)), which can be considered as a sparser version of GCN's filter.

Another set of GNNs consider using a learnable graph convolutional filter. For example, Xu et al. (2019) and Chiang et al. (2019) both propose to use $\mathcal{F}(\mathcal{G}) = \boldsymbol{A} + \epsilon\boldsymbol{I}$ where $\epsilon$ is a learnable parameter to augment self-loop skip connection. Graph Attention Networks(Velickovic et al., 2018) proposes to assign attention weight to different nodes in a neighborhood, which can be considered as a flexible learnable graph convolutional filter. Their graph filters applied on a feature matrix $\boldsymbol{X}$ can be considered as: $\forall i, j, \mathcal{F}(\mathcal{G})_{ij} = \frac{exp(\sigma(\boldsymbol{\alpha}[\boldsymbol{W}\boldsymbol{X}_i||\boldsymbol{W}\boldsymbol{X}_j]))}{\sum_{k \in \mathcal{N}_i} exp(\sigma(\boldsymbol{\alpha}[\boldsymbol{W}\boldsymbol{X}_i||\boldsymbol{W}\boldsymbol{X}_k]))}\boldsymbol{A}_{ij}$, where $\mathcal{N}_i$ is the neighborhood of node $i$, $\alpha$ is a learnable weight vector, and $||$ indicates concatenation.

## 3 ASSESSMENT TOOL FOR ANALYZING GRAPH CONVOLUTIONAL FILTERS

In this section, we introduce a novel assessment tool for analyzing graph convolutional filters. We first review the Fisher score, which is widely used in Linear Discriminant Analysis to quantify the linear separability of two sets of features. With the Fisher score, we propose the Graph Filter Discriminant Score metric to evaluate the graph convolutional filter on how well it can separate nodes in different classes.

### 3.1 THE ASSESSMENT TOOL: GRAPH FILTER DISCRIMINANT SCORE

**Fisher Score.** When coming to non-graph data, the Fisher Score (Fisher, 1936) is used to assess the linear separability between two classes. Given two classes of features $\boldsymbol{X}^{(i)}$ and $\boldsymbol{X}^{(j)}$, the Fisher Score is defined as the ratio of the variance between the classes (inter-class distance) to the variance within the classes (inner-class distance) under the best linear projection $\boldsymbol{w}$ of the original feature:

$$J(\boldsymbol{X}^{(i)}, \boldsymbol{X}^{(j)}) = \max_{\boldsymbol{w} \in \mathbb{R}^d} \frac{(\boldsymbol{w}^\top(\boldsymbol{\mu}^{(i)} - \boldsymbol{\mu}^{(j)}))^2}{\boldsymbol{w}^\top(\boldsymbol{\Sigma}^{(i)} + \boldsymbol{\Sigma}^{(j)})\boldsymbol{w}} \tag{2}$$

where $\boldsymbol{\mu}^{(i)}$ and $\boldsymbol{\mu}^{(j)}$ denotes the mean vector of $\boldsymbol{X}^{(i)}$ and $\boldsymbol{X}^{(j)}$ respectively, $\boldsymbol{\Sigma}^{(i)}$ and $\boldsymbol{\Sigma}^{(j)}$ denotes the variance of $\boldsymbol{X}^{(i)}$ and $\boldsymbol{X}^{(j)}$ respectively, and $\boldsymbol{w}$ denotes the linear projection vector which we can understand as a rotation of the coordinate system, and the $\max_{\boldsymbol{w}}$ operation is to find the best direction in which these two class of nodes are most separable. As the numerator of $J$ indicates inter-class distance and the denominator of $J$ indicates inner-class distance a larger value of $J$ indicates higher separability. Note that for given features, we can directly get the closed form solution of the optimal $\boldsymbol{w}$, with which Fisher Score could be deformed as: $J(\boldsymbol{X}^{(i)}, \boldsymbol{X}^{(j)}) = (\boldsymbol{\mu}^{(i)} - \boldsymbol{\mu}^{(j)})^\top(\boldsymbol{\Sigma}^{(i)} + \boldsymbol{\Sigma}^{(j)})^{-1}(\boldsymbol{\mu}^{(i)} - \boldsymbol{\mu}^{(j)})$. The detailed proof is provided in appendix A.2.

**Graph Filter Discriminant Score.** As mentioned before, the key component that empowers GNNs is the graph convolutional filter $\mathcal{F}(\mathcal{G})$. Intuitively, an effective filter should make the representations of nodes in different classes more separable. Therefore, we propose to use Fisher Scores of the node representations before and after applying the graph convolutional filter in order to evaluate this filter. For each pair of classes $(i, j)$, we define their Fisher Difference as $FD(i, j) = J\big((\mathcal{F}(\mathcal{G})\boldsymbol{X})^{(i)}, (\mathcal{F}(\mathcal{G})\boldsymbol{X})^{(j)}\big) - J(\boldsymbol{X}^{(i)}, \boldsymbol{X}^{(j)})$, which is the difference of their Fisher Score of representations after applying the filter $\mathcal{F}(\mathcal{G})$ and their Fisher Score of initial representations. We then define the GFD Score for the filter $\mathcal{F}(\mathcal{G})$ with respect to feature matrix $\boldsymbol{X}$ as follows:

$$GFD\big(\mathcal{F}(\mathcal{G}), \boldsymbol{X}\big) = \sum_{i \neq j} \beta_{ij} FD(i, j), \quad \beta_{ij} = \frac{n_i + n_j}{\sum_{k \neq t}(n_k + n_t)}$$

where $n_c$ is the number of nodes in class $c$. Note that the GFD Score is a weighted sum of the Fisher Difference for each pair of classes. Intuitively, the larger the GFD score, the more effective is this corresponding filter to increase the separability of node features.

The Fisher Score can be extended to evaluate non-linearly separable data in addition to linearly separable data. We claim the rationale of such measure by showing that the graph convolution can actually help non-linearly separable data to be linearly separable if the graph filter is chosen properly for a given graph. As shown in Figure 2(a)~(d), if we use a proper filter, the convolutional operation can transform three circular distributions, which are non-linearly separable, into three linearly separable clusters. Moreover, as shown in Figure 2(e)~(h), even if the original features of different classes are sampled from the same distribution, the proper graph convolutional filter can help to linearly separate the data. This phenomenon shows that if the graph structure ($\boldsymbol{A}$) is correlated with the task ($\boldsymbol{Y}$), a proper filter alone is powerful enough to empower GNNs with non-linearity, without any non-linear activation. This phenomenon is also supported by the promising result of SGC (Wu et al., 2019a), which removes all the non-linear activations in the GCN architecture. Therefore, we claim that the proposed GFD is a reasonable metric to evaluate a graph filter's effectiveness, and a good graph filter for a given graph should have a higher GFD score on that graph.

### 3.2 ASSESSING EXISTING GRAPH CONVOLUTIONAL FILTERS

With the help of the assessment tool, we now examine existing filters and try to answer the two fundamental questions: *(1) Is there a best filter that works for all graphs? (2) If not, what are the properties of graph data that will influence the performance of graph convolutional filters?*

The GFD Score we introduced in the above section can be applied to any filter on any given graph. From Table 3, we can see that most of the current GNNs fall into the following filter family: $\{(\hat{\boldsymbol{A}})^k\}$, where the base $\hat{\boldsymbol{A}}$ is a normalized adjacency matrix, and $k$ is the order of the filter. Note that there are some other variants of GNN filters that do not fall into this family, for example, GAT, but the analysis is similar. Without loss of generality, we focus on analyzing this filter family. The two main components of this filter family are the normalization strategy ($\hat{\boldsymbol{A}}$) and the order to use ($k$). For

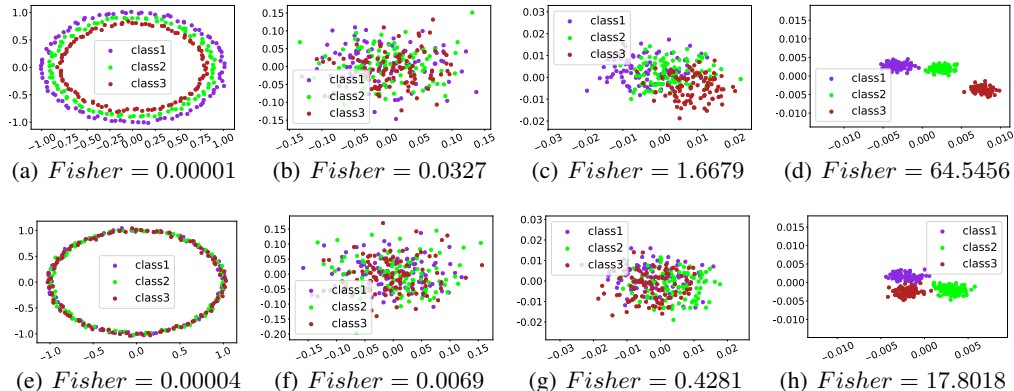

Figure 2: Each row corresponds to a graph. The i-th column corresponds to the feature distribution after applying filter $(\tilde{D}^{-1/2}\tilde{A}\tilde{D}^{-1/2})^{i-1}$. Both graphs include three classes of same size and has structure generated by SBM ($p = 0.6$, $q = 0.03$). The first graph's feature follows a circular distribution with radius $= 1, 0.9, 0.8$ and Gaussian noise $= 0.02$ for each class. The second graph's feature follows a circular distribution with radius $= 1$ and Gaussian noise $= 0.02$ for all classes.

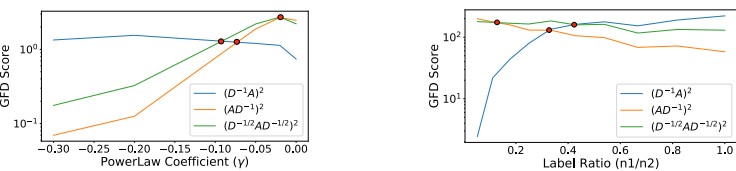

Figure 3: How power law coefficient and label ratio influence the optimal choice of filter's normalization strategy. Detailed Parameters of graph generator are provided in Appendix A.3.

simplicity, we study the roles of these two components separately, using our assessing tool to show whether there exists an optimal choice of filter for different graph data. If an optimal choice does not exist, we determine the factors that will influence our choice of component.

Through the analysis, we choose SBM and DCSBM introduced previously to generate the structures of synthetic graphs, and choose multivariate Gaussian distributions to generate features of synthetic graphs. We focusing on the structure properties that influence the optimal choice of filter. We enumerate the hyper-parameters to generate graphs with different structure properties, including the power law coefficient ($\gamma$) that controls the power law degree distribution of the graph, label ratio ($\frac{n_1}{n_2}$) that indicates how balance are the classes of this graph, density ($\frac{p+q}{2}$) that indicates the overall connectivity of the graph, and density gap ($|p-q|$) that indicates structural separability of the graph. As these properties are significant for real-world graphs, our generated synthetic graphs can cover a large range of possible graph properties, and are representative for analyzing different filters.

**Analyzing Filter's Normalization Strategy.** We consider three normalization strategies, including row normalization $D^{-1}A$, column normalization $AD^{-1}$, and symmetric normalization $D^{-1/2}AD^{-1/2}$. We calculate GFD scores of these three graph filters for graphs generated with different parameters. As shown in Figure 3, *no single normalization strategy is optimal for all graphs.*[1] Here we give an empirical explanation to this phenomenon. Note that, with the same order, each filter has the same receptive field, and different normalization strategies affect only on how to assign weights to the neighboring nodes. The row normalization strategy simply takes the mean of features of the node's neighbors. Clearly, this would help to keep every node's new representations in the same range. On the contrary, column normalization and symmetric normalization, might keep a larger representation for higher-degree nodes. Using a column-normalized adjacency matrix as the base of the graph convolutional filter is similar to the PageRank algorithm. While a node propagates its features to neighbors, this normalization strategy takes its degree into account. Thus, column normalization can be helpful when the when node degree plays an important role for classification.

---

[1]We also provide examples in Appendix A.3 to illustrate that each normalization and order will outperform others in some specific graph data, and none of a single normalization or order can be the best choice.

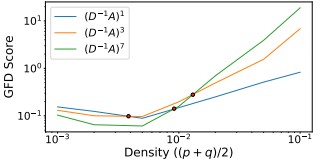 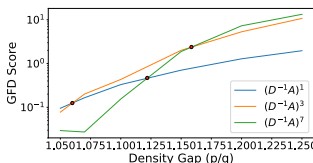

Figure 4: How density and density gap influence the optimal choice of filter's order. Detailed Parameters of graph generator are provided in Appendix A.3.

Symmetric normalization combines the properties from both the row normalization and the column normalization. Even in the case where row normalization and column normalization do not perform well, symmetric normalization still leads to promising performance.

We now examine which graph properties influence our choice of the optimal normalization strategy, which may vary per graph. We find that power law coefficient $\gamma$ is an important factor that influences the choice of normalization. As shown in Figure 3, when power-law coefficient $\gamma$ decreases (graph's extent of power-law grows), row normalization tends to have better performance. This is because row normalization helps to keep node representations in the same range, so that large representations of high degree nodes can be avoided. Therefore, it prevents nodes with similar degrees getting closer to each other and messing the classification tasks where node degrees are not important.

We also find that the label ratio ($\frac{n_1}{n_2}$) matters. As shown in Figure 3, when the size of each class becomes more imbalanced, column normalization tends to work better. This is because column normalization better leverages degree property during representation smoothing, as nodes in large-size classes tend to have larger representation since they are more likely to have higher degree. This can help nodes within different classes become more separable.

**Analyzing Filter's Order.** We then analyze what would be the best order for filters. With a high-order filter, a node can obtain information from its further neighbors, and thus the amount of information it receives during the feature propagation increases. But do we always need more information under any circumstances? The answer is no. Still, we find that, *for different graphs, the order that results in the best performance would be different*[1].

Since there is no best filter order for all the cases, we explore the factors that can influence the choice of order. We find that the density of graph and density gap between two classes have a big impact. As shown in Figure 4, when the density or density gap increases, the filter with higher order tends to be a better choice. We provide an intuitive explanation for this phenomenon as follows. Note that the feature propagation scheme is based on the assumption that nodes in the same class have a closer connection. On one hand, when the density increases, the connections between nodes are closer. Therefore, high-order filters can help gather richer information and thus reduce the variance of the obtained new node representations, so it helps nodes in the same class get smoother representations. On the other hand, when the density gap decreases, for a node, the size of neighbors within the same class becomes similar to the size of neighbors within different classes. Thus conducting high-order graph convolution operations will mix the representations of all nodes regardless of classes, which will make node classification more difficult.

## 4    LEARNING TO FIND THE OPTIMAL GRAPH CONVOLUTIONAL FILTER

Based on previous analysis, we now answer the last question: *Can we design an algorithm to adaptively find the appropriate filter for a given graph?* We develop a simple but powerful model, the Adaptive Filter Graph Neural Network (AFGNN). For a given graph, AFGNNs can learn to combine an effective filter from a set of filter bases, guided by GDF Scores.

**Adaptive Filter Graph Neural Network (AFGNN).** For simplicity, we only consider finding the optimal filter for one family of graph convolutional filters: $\mathbb{F}(\mathcal{G}) = \{\boldsymbol{I}, \tilde{\boldsymbol{D}}^{-1/2}\tilde{\boldsymbol{A}}\tilde{\boldsymbol{D}}^{-1/2}, \cdots, (\tilde{\boldsymbol{D}}^{-1/2}\tilde{\boldsymbol{A}}\tilde{\boldsymbol{D}}^{-1/2})^k, \tilde{\boldsymbol{D}}^{-1}\tilde{\boldsymbol{A}}, \cdots, (\tilde{\boldsymbol{D}}^{-1}\tilde{\boldsymbol{A}})^k, \tilde{\boldsymbol{A}}\tilde{\boldsymbol{D}}^{-1}, \cdots, (\tilde{\boldsymbol{A}}\tilde{\boldsymbol{D}}^{-1})^k\}$, where $k$ is the maximum order. Note that, we also include the identity matrix, which serves as a skip-connection, to maintain the original feature representation. Based on our previous analysis, for graphs that are not closely correlated to tasks (i.e., small density gap in SBM), the identity matrix will outperform all the other convolutional filters. We denote the above $3k + 1$ filters as

$\mathcal{F}_1^{base}(\mathcal{G}), \cdots, \mathcal{F}_{3k+1}^{base}(\mathcal{G})$, the $l$-th layer of AFGNN is defined as a learnable linear combination of these filter bases:

$$\mathcal{F}_{AFGNN}(\mathcal{G})^{(l)} = \sum_{i=1}^{3k+1} \alpha_i^{(l)} \mathcal{F}_i^{base}(\mathcal{G}), \text{ where } \alpha_i^{(l)} = \frac{\exp(\psi_i^{(l)})}{\sum_{j=1}^{3k+1} \exp(\psi_j^{(l)})} \qquad (3)$$

where $\psi^{(l)}$ is the learnable vector to combine base filters and $\alpha^{(l)}$ is its softmax-normalized version.

Comparing to GNNs with fixed filters such as GCN and SGC, our proposed AFGNN can adaptively learn a filter based on any given graph. As we have shown that no single fixed filter can perform optimally for all graphs, we conclude that an adaptive filter has more capacity to learn better representations. Comparing to other GNNs with learnable filters such as GAT, AFGNN is computationally cheaper and achieves similar or better performance on most existing benchmarks and our synthetic datasets (as shown in the experiment section). We leave expanding the base filter family and adding more complex filters such as GAT into our filter bases as future work.

**Training Loss.** To train this AFGNN model, we can simply optimize the whole model via any downstream tasks, i.e., node classification. However, as most of the semi-supervised node classification datasets only contain limited training data, the enlarged filter space will make the model prone to over-fitting. Thus, we decide to add the GFD Score as an loss term into the training loss to guide the optimization of filter weights, i.e., $\psi^{(l)}$ and to prevent overfitting:

$$\mathcal{L} = \mathcal{L}_{CE} + \lambda \mathcal{L}_{GFD}, \text{ where } \mathcal{L}_{GFD} = -\sum_{l}^{L} GFD\Big(\mathcal{F}_{AFGNN}(\mathcal{G})^{(l)}, H^{(l-1)}\Big) \qquad (4)$$

where $\mathcal{L}_{CE}$ is the cross-entropy loss of the node classification, and $\mathcal{L}_{GFD}$ is defined as the cumulative negation of GFD Score for the learned adaptive filter $\mathcal{F}_{AFGNN}(\mathcal{G})^{(l)}$ at each layer with respect to its input representation $H^{(l-1)}$. During the training, we minimize $\mathcal{L}$ to learn the proper model.

With a different choice of the weight $\lambda$ for GFD loss, we can categorize our model into:
**AFGNN$_0$**: With $\lambda = 0$, the model is only trained by $\mathcal{L}_{CE}$, which might be prone to over-fitting when data is not sufficient.
**AFGNN$_1$**: With $\lambda = 1$, the model is trained by both $\mathcal{L}_{CE}$ and $\mathcal{L}_{GFD}$ simultaneously.
**AFGNN$_\infty$**: This case is not exactly $\lambda = \infty$, and the training process is different from the other two cases. We implement the training iteratively: we optimize the combination of base filters by training only with GFD loss $\mathcal{L}_{GFD}$, then we optimize the linear transofrmation parameter $W^l$s with classification loss $\mathcal{L}_{CE}$. Note that the input feature $H^{(0)} = X$ is invariant, we can pre-train the optimal filter for first layer and fix it.

## 5 EXPERIMENTS

**Dataset** We first evaluate AFGNN on three widely used benchmark datasets: Cora, Citeseer, and Pubmed (Sen et al., 2008). As these datasets are not sensitive enough to differentiate the models, we need more powerful datasets that can evaluate the pros and cons of each model. Based on our findings in section 3.2, we generate two synthetic benchmarks called SmallGap and SmallRatio. SmallGap corresponds to the case in which the density gap of the graph is close to 1. This indicates that the graph structure does not correlate much to the task, thus $I$ would be the best filter in this case. SmallRatio corresponds to the case in which the label ratio is small, i.e. the size of one class is clearly smaller than the other, and column normalization $AD^{-1}$ is the best normalization[2].

**Baselines and Settings.** We compare against 5 baselines, including GCN, GIN, SGC, GFNN, and GAT. To make fair comparisons, for all the baseline GNNs, we set the number of layers (or orders) to be 2, and tune the parameters including learning rate, weight decay, and number of epochs[3]. For all the benchmark datasets, we follow the data split convention[2]. For the synthetic dataset, we conduct 5-fold cross-validation, randomly split the nodes into 5 groups of the same size, take one group as the training set, one as the validation set and the remaining three as the test set. Each time we pick

---

[2]The statistics of benchmark and parameters used to generate synthetic dataset are in Appendix A.6.
[3]The details about baseline code and hyperparameters settings are provided in Appendix A.7

| Dataset | GCN | GIN | SGC | GFNN | GAT | $\text{AFGNN}_0$ | $\text{AFGNN}_1$ | $\text{AFGNN}_\infty$ |
|---|---|---|---|---|---|---|---|---|
| Cora | 80.85±0.43 | 76.37±0.75 | 81.14±0.05 | 80.42±0.70 | **82.90± 0.01** | 61.22±1.61 | 81.16±0.48 | **81.40 ±0.03** |
| Citeseer | 71.19±0.60 | 67.85±0.52 | 71.91±0.01 | 71.15±0.55 | **72.20±0.07** | 60.58±0.91 | 71.32±0.85 | **71.80±0.01** |
| Pubmed | 79.08±0.23 | 74.23±1.76 | 78.50±0.00 | **79.12±0.23** | 78.50±0.01 | 72.58±2.84 | 78.51±0.14 | **79.20±0.01** |
| SmallGap | 82.78±0.20 | 76.83±0.87 | 74.53±0.94 | 83.38±0.30 | 85.26±0.07 | 90.85±3.24 | **99.91±0.04** | **99.95±0.01** |
| SmallRatio | **87.79±1.05** | 77.82±3.40 | 87.14±0.19 | 83.75±0.20 | 82.10±0.01 | 74.45±4.81 | 85.69±3.69 | **93.80±1.11** |

Table 1: Test accuracy of different models on both benchmark and synthetic datasets.

| Graph Filters | Cora | Citeseer | Pubmed | SmallGap | SmallRatio |
|---|---|---|---|---|---|
| $I$ | 13.15 | 25.05 | 11.75 | **34.13** | 3.62 |
| $\tilde{D}^{-1}\tilde{A}$ | 33.68 | 48.24 | 22.05 | 0.97 | 6.25 |
| $(\tilde{D}^{-1}\tilde{A})^2$ | **58.48** | **67.53** | **37.70** | 18.17 | 10.50 |
| $\tilde{A}\tilde{D}^{-1}$ | 28.35 | 43.99 | 12.44 | 0.97 | 10.67 |
| $(\tilde{A}\tilde{D}^{-1})^2$ | 49.64 | 62.16 | 22.74 | 16.27 | **80.37** |
| $\tilde{D}^{-1/2}\tilde{A}\tilde{D}^{-1/2}$ | 31.87 | 47.20 | 20.09 | 0.97 | 8.51 |
| $(\tilde{D}^{-1/2}\tilde{A}\tilde{D}^{-1/2})^2$ | 54.76 | 65.50 | 34.61 | 17.58 | 49.69 |
| $\text{AFGNN}_0$ | 19.72 | 34.84 | 12.02 | 34.29 | 8.50 |
| $\text{AFGNN}_1$ | 58.28 | **67.83** | **38.61** | 34.59 | 77.81 |
| $\text{AFGNN}_\infty$ | **58.48** | 67.68 | **38.61** | 34.68 | 80.37 |

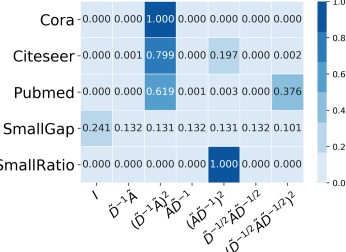

Table 2: Fisher Score after appying different filters on both benchmark and synthetic datasets

Figure 5: Base filter combination Learned by $\text{AFGNN}_\infty$

the model with highest validation accuracy and record its test accuracy. For each dataset, we run the experiment 10 times and compute the mean and standard deviation of recorded test accuracy.

**Classification Performance.** As is shown in Table 1, our proposed $\text{AFGNN}_\infty$ model can consistently achieve competitive test accuracy. On Pubmed, SmallGap, and SmallRatio, $\text{AFGNN}_\infty$ can achieve the best results among all the baseline models. On Cora and Citeseer, though GAT outperforms our proposed model a little bit, however, as shown in Table 6,7, GAT takes a longer time to train and converge, and has more memory cost as well. Also, when the given graph is simple, GAT would suffer unavoidable overfitting problem. We further compare our $\text{AFGNN}_0$, $\text{AFGNN}_1$, $\text{AFGNN}_\infty$ to examine the role of GFD loss. The $\text{AFGNN}_0$ performs quite poorly on all the datasets, implying that the larger search space of the filter without GFD loss is prone to over-fitting, while $\text{AFGNN}_1$ and $\text{AFGNN}_\infty$ perform much better. Also, $\text{AGFNN}_\infty$ has superior performance compared to $\text{AFGNN}_1$, which indicates the GFD Score is indeed a very powerful assessment tool.

**Graph Filter Discriminant Analysis.** We are also interested to see whether the proposed method can indeed learn the best combination of filters from the base filter family. To do so, we calculate the GFD Score of the first-layer filter learned by $\text{AFGNN}_0$, $\text{AFGNN}_1$, $\text{AFGNN}_\infty$ and the seven base filters on the test set for each dataset. For the AFGNN models, the filter is trained with the training set for each dataset. Table 2[4] and Figure 5 show the results, we can see that our proposed method can indeed learn a combined filter on all the datasets. Specifically, in all the benchmark datasets, the best base filter is $(\tilde{D}^{-1}\tilde{A})^2$, and our proposed adaptive filter not only picks out the best base filter but also learns a better combination. For the two synthetic datasets, where $I$ and $(\tilde{A}\tilde{D}^{-1})^2$ are the best filters, our algorithm can also learn to pick them out. We thereby conclude that the proposed GFD loss can help find an appropriate filter for a given dataset.

## 6 CONCLUSION

Understanding the graph convolutional filters in GNNs is very important, as it can help to determine whether a GNN will work on a given graph, and can provide important guidance for GNN design. In our paper, we focus on the semi-supervised node classification task. We first propose the Graph Filter Discriminant Score as an assessment tool for graph convolutional filter evaluation, and then apply this GFD Score to analyze a family of existing filters as a case study. Using this tool, we learn that no single fixed filter can produce optimal results on all graphs. We then develop a simple but powerful GNN model: Adapative Filter Graph Neural Network, which can learn to combine a

---

[4]Note that fisher score before applying filter is a fixed value, therefore, to compare GFD of each filter, we only need to compare Fisher Score after appying different filters.

family of filters and obtain a task-specific powerful filter. We also propose to add the negative GFD Score as an extra component to the objective function, it can act as a guidance for the model to learn a more effective filter. Experiments show that our approach outperforms many existing GNNs on both benchmark and synthetic graphs.

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

| GNNs | Graph Convolutional Filters |
|------|------------------------------|
| GCN (Kipf & Welling, 2017) | $\mathcal{F}(\mathcal{G}) = \tilde{D}^{-1/2}\tilde{A}\tilde{D}^{-1/2}$ |
| SGC (Wu et al., 2019a) | $\mathcal{F}(\mathcal{G}) = (\tilde{D}^{-1/2}\tilde{A}\tilde{D}^{-1/2})^k$ |
| GFNN (Maehara, 2019) | $\mathcal{F}(\mathcal{G}) = (\tilde{D}^{-1/2}\tilde{A}\tilde{D}^{-1/2})^k$ |
| GIN (Xu et al., 2019) | $\mathcal{F}(\mathcal{G}) = A + \epsilon I$ |
| GAT (Velickovic et al., 2018) | $\mathcal{F}(\mathcal{G}) = Q$, where $Q$ is parametric attention function of $X$ and $A$ |

Table 3: A Summary of Graph Filters of Existing GNNs.

Felix Wu, Tianyi Zhang, Amauri Holanda de Souza Jr, Christopher Fifty, Tao Yu, and Kilian Q Weinberger. Simplifying graph convolutional networks. arXiv preprint arXiv:1902.07153, 2019a.

Zonghan Wu, Shirui Pan, Fengwen Chen, Guodong Long, Chengqi Zhang, and Philip S Yu. A comprehensive survey on graph neural networks. arXiv preprint arXiv:1901.00596, 2019b.

Keyulu Xu, Weihua Hu, Jure Leskovec, and Stefanie Jegelka. How powerful are graph neural networks? In 7th International Conference on Learning Representations, ICLR 2019, New Orleans, LA, USA, May 6-9, 2019, 2019. URL https://openreview.net/forum?id=ryGs6iA5Km.

Rex Ying, Ruining He, Kaifeng Chen, Pong Eksombatchai, William L Hamilton, and Jure Leskovec. Graph convolutional neural networks for web-scale recommender systems. In Proceedings of the 24th ACM SIGKDD International Conference on Knowledge Discovery & Data Mining, pp. 974–983. ACM, 2018.

Jie Zhou, Ganqu Cui, Zhengyan Zhang, Cheng Yang, Zhiyuan Liu, and Maosong Sun. Graph neural networks: A review of methods and applications. arXiv preprint arXiv:1812.08434, 2018.

# A    APPENDIX

## A.1    SUMMARY OF GRAPH FILTERS FOR EXISTING GNNS.

Table 3 summarized the graph filters for existing GNNs.

## A.2    PROOF OF PROPOSITION 1

**Proof** According to the conclusions in linear discriminant analysis, the maximum separation occurs when $w \propto (\Sigma^{(i)} + \Sigma^{(j)})^{-1}(\mu^{(i)} - \mu^{(j)})$. Note that, when we want to apply this fisher linear discriminant score in our problem, the linear transformation part in our classifier (and also the linear transformation part in GNN) will help to find the best $w$. Thus, we can directly plug the optimum solution $w^* = c(\Sigma^{(i)} + \Sigma^{(j)})^{-1}(\mu^{(i)} - \mu^{(j)})$ into this formula, here $c$ is a scalar. Then, we'll have:

$$
\begin{aligned}
J(X^{(i)}, X^{(j)}) &= \max_{w \in \mathbb{R}^d} \frac{(w^\top(\mu^{(i)} - \mu^{(j)}))^2}{w^\top(\Sigma^{(i)} + \Sigma^{(j)})w} \\
&= \frac{((c(\Sigma^{(i)} + \Sigma^{(j)})^{-1}(\mu^{(i)} - \mu^{(j)}))^\top(\mu^{(i)} - \mu^{(j)}))^2}{(c(\Sigma^{(i)} + \Sigma^{(j)})^{-1}(\mu^{(i)} - \mu^{(j)}))^\top(\Sigma^{(i)} + \Sigma^{(j)})(c(\Sigma^{(i)} + \Sigma^{(j)})^{-1}(\mu^{(i)} - \mu^{(j)}))} \\
&= \frac{((\mu^{(i)} - \mu^{(j)})^\top(\Sigma^{(i)} + \Sigma^{(j)})^{-1}(\mu^{(i)} - \mu^{(j)}))^2}{(\mu^{(i)} - \mu^{(j)})^\top(\Sigma^{(i)} + \Sigma^{(j)})^{-1}(\mu^{(i)} - \mu^{(j)})} \\
&= (\mu^{(i)} - \mu^{(j)})^\top(\Sigma^{(i)} + \Sigma^{(j)})^{-1}(\mu^{(i)} - \mu^{(j)})
\end{aligned}
\tag{5}
$$

Thus we completed the proof. ∎

### A.3   No Best Filter for All Graphs

#### A.3.1   Examples of "No Best Normalization Strategy for All"

Figure 6 provides two examples to show there is no best normalization strategy for all graphs. For both examples, we fix the order of filter to be 2.

The first row shows a case in which row normalization is better than the other two. The corresponding graph contains 2 classes of nodes with size 500. The graph structure is generated by DCSBM with $p = 0.3$, $q = 0.05$, power law coefficient $\gamma = -0.9$. The features for two classes satisfy multivariate distribution with an identity co-variance matrix, and with mean (0.2,0.2) and (0,0) respectively. In this example, we can clearly see that with other two normalization strategy, some high-degree hubs show up in the upper right corner from both class, which is harmful for classification. We generate this example to illustrate the benefit of row normalization because row normalization would be very helpful for a graph with power law degree distribution, which contains some nodes with unusually large degree (those nodes are called hubs), since it can help avoid those hubs obtaining larger representations and thus be mis-classified.

The second row shows a case in which column normalization is better than the other two. The corresponding graph contains 2 classes of nodes with size 900 and 100 respectively. The graph structure is generated by SBM with $p = 0.3$, $q = 0.2$. The features for two classes satisfy multivariate distribution with an identity co-variance matrix, and with mean (-0.2,-0.2) and (0.2,0.2) respectively. We generate this example to illustrate the benefit of column normalization because under this case, we should consider taking more degree information into consideration. Therefore, column normalization would be more helpful.

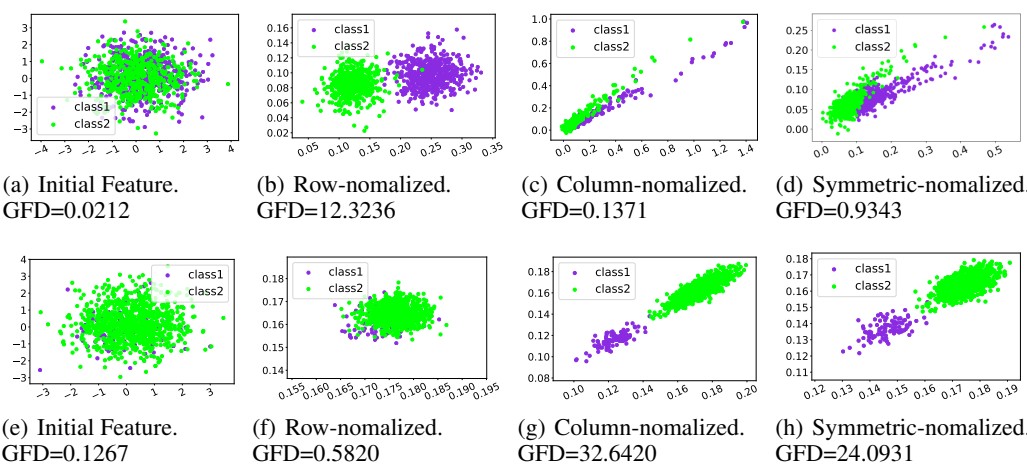

(a) Initial Feature.
GFD=0.0212

(b) Row-nomalized.
GFD=12.3236

(c) Column-nomalized.
GFD=0.1371

(d) Symmetric-nomalized.
GFD=0.9343

(e) Initial Feature.
GFD=0.1267

(f) Row-normalized.
GFD=0.5820

(g) Column-normalized.
GFD=32.6420

(h) Symmetric-normalized.
GFD=24.0931

Figure 6:  Examples of "No Best Normalization Strategy for All"

#### A.3.2   Examples of "No Best Order for All"

Figure 7 provides two examples to show there is no best order for all graphs. For both examples, we fix the normalization strategy to be row normalization, and varies order to be 2, 4, 6.

The first row shows a case in which small order is better than the large ones. The corresponding graph contains 2 classes of nodes with same size 500. The graph structure is generated by SBM with $p = 0.215$, $q = 0.2$. The features for two classes satisfy multivariate distribution with an identity co-variance matrix, and with mean (0.5,0.5) and (0,0) respectively.

The second row shows a case in which large order is better than the smaller ones. The corresponding graph contains 2 classes of nodes with same size 500. The graph structure is generated by SBM with $p = 0.75$, $q = 0.6$. The features for two classes satisfy multivariate distribution with an identity co-variance matrix, and with mean (0.5,0.5) and (0,0) respectively.

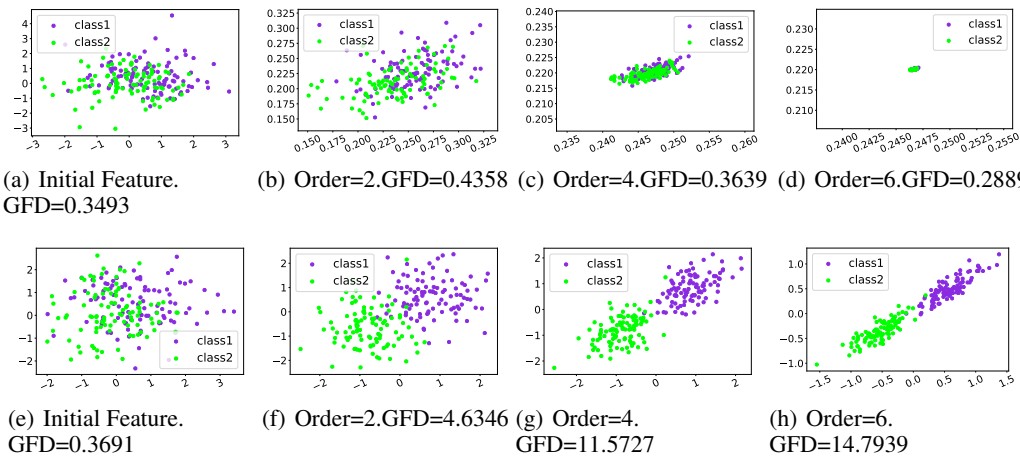

(a) Initial Feature. GFD=0.3493

(b) Order=2.GFD=0.4358

(c) Order=4.GFD=0.3639

(d) Order=6.GFD=0.2889

(e) Initial Feature. GFD=0.3691

(f) Order=2.GFD=4.6346

(g) Order=4. GFD=11.5727

(h) Order=6. GFD=14.7939

Figure 7: Examples of "No Best Order for All"

### A.3.3    ILLUSTRATION OF GRAPH GENERATOR FOR CURVES IN SECTION 3.2

For the curves indicating how powerlaw coefficient influence the choice of normalization in Figure 3, we generate the corresponding graphs structure by DCSBM with fixed $p = 0.3$, $q = 0.2$ and varies the powerlaw coefficient from -0.3 to 0. The graph contains two classes of nodes, and is of size 400 and 600 for each class respectively. The feature for each class satisfies multivariate normal distribution with identity co-variance matrix, and with mean (0,0) and (0.2,0.2).

For the curves indicating how label ratio influence the choice of normalization in Figure 3, we generate the corresponding graphs structure by SBM with fixed $p = 0.3$, $q = 0.1$ and varies the label ratio. The graph contains a total number of 1000 nodes in two classes. The feature for each class satisfies multivariate normal distribution with identity co-variance matrix, and with mean (0,0) and (0.5,0.5).

For the curves indicating how density influence the choice of normalization in Figure 4, we generate the corresponding graphs structure by SBM with fixed density gap $p/q = 1.5$ and varies the density by varying q. The graph contains two classes of node of size 500. The feature for each class satisfies multivariate normal distribution with identity co-variance matrix, and with mean (0,0) and (0.5,0.5).

For the curves indicating how density gap influence the choice of normalization in Figure 4, we generate the corresponding graphs structure by SBM with fixed density $p + q = 0.6$ and varies the density gap. The graph contains two classes of node of size 500. The feature for each class satisfies multivariate normal distribution with identity co-variance matrix, and with mean (-0.2,-0.2) and (0.2,0.2).

### A.4    FLOWCHART OF AFGNN FOR NODE CLASSIFICATION

The following flowchart (Figure 8) describes the process of how a one-layer AFGNN tackle node classification task.

### A.5    FEATURE PROPAGATION USING DIFFERENT FILTERS

We reduced the dimension of feature by t-SNE (Maaten & Hinton (2008)). We annotate the filter and the GFD Score in title of each subfigure. Note that, identity also corresponds to the initial feature. The figure is the feature representation obtained after conduct graph convolution operation once with the corresponding filter.

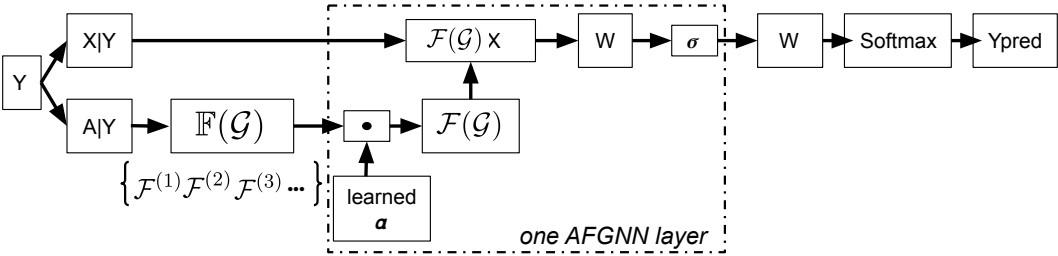

Figure 8: Flowchart of a 1-layer AFGNN for node classification.

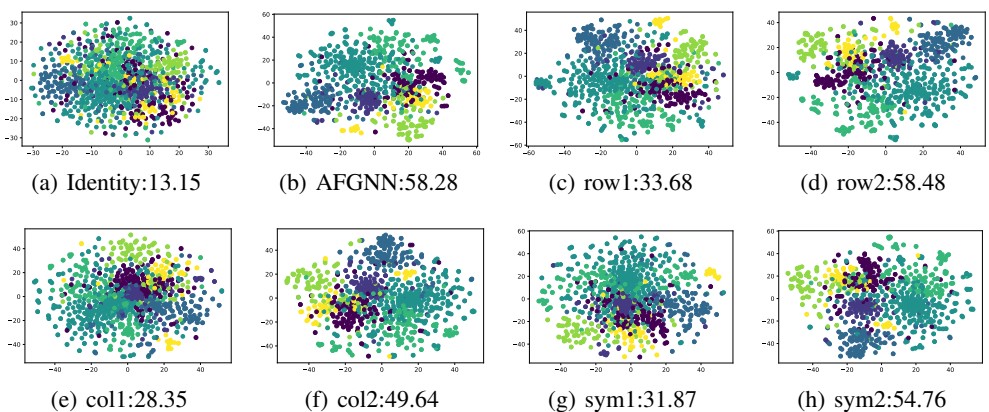

(a) Identity:13.15  (b) AFGNN:58.28  (c) row1:33.68  (d) row2:58.48

(e) col1:28.35  (f) col2:49.64  (g) sym1:31.87  (h) sym2:54.76

Figure 9: Feature Visualization

## A.6 DATASET

### A.6.1 BENCHMARK DATASET

We use three benchmark dataset: Cora, Citeseer and Pubmed for the node classification task. Their statictics are in table4. Beside number of nodes, edges, classes, the dimension of feature, and the data split strategy, we also show the *class ratio variance*, which can indicates if this dataset is imbalance or not, *density gap*, which indicates the dependency of structure and labels, *and density*, which indicates the overall connectivity of a graph. We provide the degree distribution in Figure 10, and we can clearly find that these benchmark datasets has power law degree distribution.

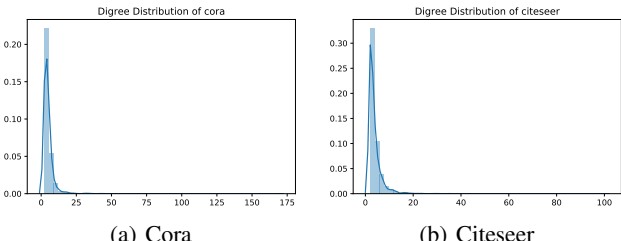

(a) Cora  (b) Citeseer

Figure 10: Degree Distribution of Benchmark Dataset

| Dataset | **Cora** | **Citeseer** | **Pubmed** |
|---|---|---|---|
| Nodes | 2708 | 3327 | 19717 |
| Edges | 5429 | 4732 | 44338 |
| Classes | 7 | 6 | 3 |
| Feature | 1433 | 3703 | 500 |
| Train | 140 | 120 | 60 |
| Validation | 500 | 500 | 500 |
| Test | 1000 | 1000 | 1000 |
| Ratio Variance | 0.0052 | 0.0020 | 0.0079 |
| Density Gap | 0.013 | 0.0051 | 0.0006 |
| Density | 0.0068 | 0.0029 | 0.0004 |

Table 4: Statistics of Benchmark Dataset

| | Cora | Citeseer | Pubmed |
|---|---|---|---|
| GCN | 81.5 | 70.3 | 79.0 |
| GIN | − | − | − |
| SGC | 81.0±0.0 | 71.9±0.1 | 78.9±0.0 |
| GFNN | 80.9±1.3 | 69.3±1.1 | 81.2±1.5 |
| GAT | 83.0±0.7 | 72.5±0.7 | 79.0±0.3 |

Table 5: Baseline's Accuracy on Benchmark Dataset

### A.6.2 SYNTHETIC DATASET

We also generated two synthetic datasets: SmallGap and SmallRatio. For SmallGap, we use SBM to generate a two class network with $p = 0.2$ and $q = 0.199$. The density gap $p/q$ is very small in this case. They have the same number of nodes and both have 64 dimension features sampled from gaussian distributions with different mean and same variance. For SmallRatio, we use SBM to generate a two class network, which has 200 nodes for one class and 800 nodes for the other. This dataset is called SmallRatio because $n1/n2 = 0.25$ is small. Their 64 features are sampled from gaussian distributions with different mean and different variance. The detailed generation process and parameter can be found in our code.

### A.7 MODEL HYPER PARAMETERS

For GCN, SGC, GFNN, GAT, we directly use their public implementations. For GIN, the initial code is not for node classification task, so we modify their code following Xu et al. (2019) to conduct experiments.

We tune the number of epochs based on convergence performance. For learning rate and weight decay, we follows the parameter setting provides by the corresponding public implementations unless we find better parameters. The tuned parameters can be found in our code resource.

### A.8 BASELINE ACCURACY ON BENCHMARK DATASET

We report the accuracy of node classification task for baseline models on Cora, Citeseer, and Pubmed provided by corresponding literature. Since GIN (Xu et al., 2019) is not originally evaluated on node classification task, we do not have the reported number here. The results is in Table 5.

### A.9 TIME AND MEMORY COST COMPARISON

Both our AFGNN model and GAT model have a learnable filter. We provide time and memory complexity comparison on benchmark datasets here to compare these two models.

As shown in Table 6, GAT's time cost is at least three times of AFGNN's time cost on both Cora and Citeseer dataset. As shown in Table 7, AFGNN's memory cost on both Cora and Citeseer are half of GAT's memory cost. GAT does not have recorded time and memory cost for Pubmed dataset

| | Cora | | | Citeseer | | | Pubmed | | |
|---|---|---|---|---|---|---|---|---|---|
| | time | num | total | time | num | total | time | num | total |
| $AFGNN_0$ | 0.055 | 96.5 | 5.309 | 0.116 | 117.5 | 13.579 | 0.376 | 137.0 | 51.456 |
| $AFGNN_1$ | 0.086 | 129.3 | 11.1 | 0.136 | 155.4 | 21.177 | 0.379 | 136.1 | 51.62 |
| $AFGNN_\infty$(filter) | 0.106 | 53 | 5.593 | 0.146 | 48 | 7.023 | 0.377 | 200 | 75.456 |
| $AFGNN_\infty$(classification) | 0.005 | 200 | 0.914 | 0.006 | 400 | 2.293 | 0.006 | 400 | 2.246 |
| $AFGNN_\infty$(overall) | - | - | 6.507 | - | - | 9.315 | - | - | 77.702 |
| GAT | 0.156 | 382.8 | 59.625 | 0.168 | 379.3 | 63.462 | - | - | - |

Table 6: Time Cost

| | Cora | Citeseer | Pubmed |
|---|---|---|---|
| $AFGNN_0$ | 861 | 1369 | 1351 |
| $AFGNN_1$ | 863 | 1369 | 1351 |
| $AFGNN_\infty$ | 863 | 1369 | 1351 |
| GAT | 1733 | 2345 | - |

Table 7: Memory(MB) Cost

| | precision class0 | precision class1 | F1 class0 | F1 class1 | micro F1 | macro F1 |
|---|---|---|---|---|---|---|
| GCN | 95.70±1.62 | 64.50±33.80 | 97.60±0.80 | 39.60±25.97 | 95.52±1.24 | 68.64±13.24 |
| SGC | 94.30±0.45 | 21.70±33.21 | 97.00±0.00 | 9.50±14.74 | 94.34±0.36 | 53.32±7.51 |
| GFNN | **98.40±1.50** | 76.20±25.57 | **98.80±0.60** | 72.10±24.60 | **97.42±1.10** | **85.40±12.32** |
| GIN | 95.90±1.45 | 62.00±35.49 | 97.60±0.66 | 38.80±25.49 | 95.20±0.97 | 68.12±12.92 |
| GAT | 96.50±0.67 | **85.00±30.00** | 98.00±0.00 | 35.40±14.60 | 96.43±0.43 | 66.84±7.40 |
| $AFGNN_0$ | 95.00±0.77 | 53.90±39.97 | 97.20±0.40 | 19.90±19.52 | 94.62±0.68 | 58.62±9.89 |
| $AFGNN_1$ | 94.80±1.17 | 40.00±48.99 | 97.40±0.80 | 16.80±27.15 | 94.84±1.23 | 57.09±13.88 |
| $AFGNN_\infty$ | **98.20±0.40** | **83.40±0.80** | **99.00±0.00** | **77.80±1.60** | **97.38±0.18** | **88.22±0.96** |

Table 8: Performance on OAG SmallRatio Dataset

because it requires too much memory cost and is not able to run on GPU. Therefore, AFGNN needs less time and memory cost than GAT.

## A.10 SUPPLEMENTARY EXPERIMENT

We generate a real-world dataset with imbalanced classes to justify hard cases may exist in real-world datasets. We download a large scale academic graph called Open Academic Graph (OAG), and choose two fields that have a large disparity in the number of papers: (1) "History of ideas", which consists of 1041 papers; (2) "Public history", which consists of 150 papers. Obviously this two classes are imbalanced, and fall in the large label ratio gap problem. We run supplementary experiment on the generated OAG graph, the experiment setting remains the same as experiment settings for synthetic graphs.

Table 8 shows the experiment results. To evaluate the models, we compare their F1 score for each class, the weighted average F1 score (micro F1), and the average F1 score (macro F1). Our $AFGNN_\infty$ model shows superior performance on this dataset.

