# OpenReview forum: "Demystifying Graph Neural Network Via Graph Filter Assessment"
_ICLR.cc/2020/Conference — Reject_

### Official Review · AnonReviewer2 · 2019-10-24
**Official Blind Review #2**

**Rating:** 8

**Review:**

This is a very interesting study about GNN. Authors proposed to extend LDA as a discrimination evaluator for graph filleters. Also authors proposed Adaptive Filter Graph Neural Network to find the optimal filter within a limited family of graph convolutional filters. The whole study is novel, and beneficial for the community of graph neural network study. It provides a new way to understand and evaluate GNN.

There are some questions authors should clarify. And some writing errors to correct.
Eq (3) defines GFD for a pair of classes i and j. For a graph with more than two classes, the GFD will be the average of all pairs? Will class imbalance will have any impact on this GFD measure?
Errors:
•	we studies the roles of this two components
•	there exist a best choice
•	we only consider to to find


**Experience Assessment:**

I have published in this field for several years.

**Review Assessment: Checking Correctness Of Derivations And Theory:**

I assessed the sensibility of the derivations and theory.

**Review Assessment: Checking Correctness Of Experiments:**

I assessed the sensibility of the experiments.

**Review Assessment: Thoroughness In Paper Reading:**

I read the paper thoroughly.

---

> ### Author Response · Authors · 2019-11-11
> **Thank you for your support for our paper!**
>
> Thank you so much for the positive feedback! We really appreciate your support for our paper as well as your constructive suggestions. We have improved our paper based on your advice (we marked the modifications related to your suggestions with blue text, and highlighted the previous version with strikethrough):
> https://drive.google.com/file/d/1wJYwz1oPDK1-NbpesHUR6ZMCSBRVxdh3/view?usp=sharing
>
> For Eq. (3): to get the GFD score for a multi-class setting, we first calculate the Fisher Difference for each possible pair of classes, then normalize them based on class size, and finally sum them together to get GFD score. Based on the normalization, the class imbalance would not be a problem. We have improved our writing to make this part more clear.
> For the writing errors, thank you for pointing them out. We have corrected all the errors pointed out by you and also have carefully gone through the paper to improve the writing.

---

### Official Review · AnonReviewer3 · 2019-10-24
**Official Blind Review #3**

**Rating:** 1

**Review:**

In this paper, the authors raise and address three questions about graph neural network: (1) Whether there is a best filter that works for all graphs. (2) Which properties of the graph will influence the performance of the graph filter. (3) How to design a method to adaptively find the optimal filter for a given graph.
The paper proposes an assessment method called the Graph Filter Discriminant Score based on the Fisher Score. It measures how well the graph convolutional filter discriminate node representations of different classes in the graph by comparing the Fisher score before and after the filter.
Based on the GFD scores of different normalization strategy and different order of the graph filter in the experiments on synthetic data, the authors answer the first two questions: (1) There is no optimal normalization for all graphs. (2) row normalization performs better with lower power-law coefficient, but works worse with imbalanced label classes and large density gap.
For the third question, the authors propose a learnable linear combination of a limited family of graph convolutional filters as the layer of model AFGNN, which can learn the optimal arguments of the combination based on the FGD score.
The paper focuses on a significant topic and proposes an assessment tool for the graph filters. Based on that, it also introduces a model to choose filters from a family of filters for any specific graph.
The description of preliminaries is clear.
The observations of the impact of the graph properties on the filter choice are interesting and explanations are provided.
The results of the test accuracy on both bench mark and synthetic datasets demonstrate the good performance of the proposed model.
It is good that the paper provides proof for the claim that the graph convolutional can help the non-linear separable data to be linearly separable, so it is reasonable to use Fisher score. However, does this claim support the second term in equation (3), where the Fisher score is used to evaluate before the filters applied?
The presentation of the last paragraph of “graph filter discriminant score” in page 4 can be improved. The Figure references seem incorrect and confusing.
The analysis of the influence of label ratio seems not accurate enough.
For the GFD score comparison in Figure 4, why choose order 1,3,7 for density and different order 2,3,6 for density gap?
What is the meaning of the symbol psi(l)?
It would be better if the explanation of the raining loss section is more detailed and clear.
What is “AFGNN_P” in the experiment analysis?
It could be interesting to see the comparison of time between the proposed method and the GAT.
For the graph filter discriminate analysis, is it fair to compare the learned layer with the other base filter using the GFD score? Since the learned layer is picked with highest GFD score. Maybe one or two sentences on this will be helpful.
The writing of the paper must be improved. Too many typos and grammar problems will impair the presentation and the reader can be distracted.
Minor comments:
The layout of the sub caption of Figure 1 can be improved.
The usage of capital letter in the phrase “density gap” is inconsistent.
“As shown in figure” instead of “As is shown in figure”.
Many sentences miss article.
There are many typos in the writing.
For example, “Note that for given (feature)”, “…make the representation of nodes in different (class) more separable.”, “Noted that there are some other (variant) of GNN filters that (does) not fall into…” in page 4.
“Here we give (a) empirical explanation to this phenomenon”, “this normalization strategy (take) into account…”, “Thus even in the case that the other two (doesn’t) perform well…” in page 5.
“…a very important factor that (influence) the choice of normalization strategy”, “when power-law coefficient (decrease)”, “when the (sizes) of each class (become) more imbalanced”,  “This is because column normalization better (leverage) …” , “in a similar manner (with) label ratio”, “when the density or density gap (increase)”, “high-order filters can help gather… and thus (makes) the representations…”, “when the density gap (increase)” in page 6.
These can be continued but it is obvious that this paper needs proofreading.


**Experience Assessment:**

I do not know much about this area.

**Review Assessment: Checking Correctness Of Derivations And Theory:**

I assessed the sensibility of the derivations and theory.

**Review Assessment: Checking Correctness Of Experiments:**

I assessed the sensibility of the experiments.

**Review Assessment: Thoroughness In Paper Reading:**

I read the paper at least twice and used my best judgement in assessing the paper.

---

> ### Author Response · Authors · 2019-11-11
> **Response to Reviewer3**
>
> Thank you for your constructive comments! We improved our paper based on your advice (we marked the modifications related to your suggestions with red text, and highlighted the previous version with strikethrough):
> https://drive.google.com/file/d/1adtmKH61RLyBKzn46DWdEj5JQvu5M5WO/view?usp=sharing
>
> First, we want to emphasize this paper’s contribution. Our work provides a theoretical understanding to  GNNs in a novel way, and we are the first to analyze GNNs for the node classification task from a data perspective. Since rich literature demonstrated that the key of GNNs lies in their graph convolutional filters, we propose a new assessment tool (GFD) to evaluate the effectiveness of filters given a specific graph data Further, this tool is applied to analyze existing filters and found some meaningful insights. Finally, we propose the AFGNN model to automatically learn the best filter from the given family (i.e., learn the best coefficients for the linear combination of a set of filters) for the given graph data.
>
>
> Now we address your comments and concerns in detail:
>
> Q1: Is it reasonable to use the Fisher score” to support the second term in equation (3), where the Fisher score is used to evaluate before the filters applied?
> A1: We’d like to clarify our GFD is an assessment tool to see whether a filter is good for a particular graph data.
> Fisher Score is used to evaluate the separability of data. GFD defined in  Eq. (3), which is the Difference of Fisher Score before and after the filter, is to evaluate whether a filter can increase the data separability.
> As we show that a good graph convolutional filter can help the non-linear separable data to be linearly separable, we can expect the GFD score is higher for these filters. Note that not every graph filter has this property for every dataset, and our final model is to learn the best filter that could enhance this property for a given dataset. Experimental results in Figure 5 and Table 2 also support this claim.
>
> Q2: The presentation of the last paragraph of “graph filter discriminant score” in page 4 can be improved. The Figure references seem incorrect and confusing.
> A2: As mentioned in Q1, we use Fisher score to evaluate the separability of two classes, we use Fisher Score Difference to evaluate the power of a filter on two classes, and we use GFD, which is a weighted sum of Fisher Score Difference of each pair of classes, to evaluate the power of a filter on the given graph. We have revised our writing and corrected our Figure reference to make this more clear.
>
> Q3: The analysis of the influence of label ratio seems not accurate enough.
> A3: Suppose the density and density gap are fixed, when label ratio drops, which means the two classes become more imbalanced, and nodes in a larger class tend to have more neighbors. Then, with the column normalization strategy that does not have any constraint on the range of representation, those nodes with a larger degree tend to aggregate more information and thus have larger new representations. This would be helpful to differentiate the two classes. Take Figure 6 (g) as an example, nodes in the large-size class (green nodes) are gathered in the upper right part while nodes in the small-size class (purple nodes) are gathered in the lower left part, so the two classes become more separable after applying a column-normalized filter. We revised our writing to make this more clear.
>
>
> Q4: For the GFD score comparison in Figure 4, why choose order 1,3,7 for density and different order 2,3,6 for the density gap?
> A4: Thanks for pointing out. Previously we just pick the orders that can show our findings most clearly, but we agree it is important to use consistent choice in two subfigures. We now choose the same set of orders in these two figures. The result remains the same.
>
> Q5: What is the meaning of the symbol psi(l)?
> A5: It is a learnable intermediate weight (before normalization) for each base filter. We then apply softmax normalization to it to get alpha(l). We revised our writing to make this more clear.

---

> > ### Author Response · Authors · 2019-11-11
> > **Response to Reviewer3 (cont.)**
> >
> > Q6: It would be better if the explanation of the training loss section is more detailed and clear.
> > A6: Generally, our overall loss is a weighted sum of cross-entropy loss in terms of node classification and GFD loss in terms of the filter’s capability in enhancing linear separability. By changing the value of the weight, we have $AFGNN_0$, $AFGNN_1$, and $AFGNN_{infinity}$. $AFGNN_0$ and $AFGNN_1$ correspond to the case that the weight of GFD loss is 0 and 1 respectively, and parameter $\alpha$ and W are optimized simultaneously with overall loss. $AFGNN_{infinity}$ is different, and it is not exactly the case where the weight equals infinity. To train $AFGNN_{infinity}$, we iteratively optimize $\alpha$ and W with GFD loss and classification loss respectively. Thanks for pointing out this unclear part, we have revised our writing to make it more clear.
> >
> > Q7: What is “AFGNN_P” in the experiment analysis?
> > A7: It should be $AFGNN_{infinity}$, thanks for pointing out this typo.
> >
> > Q8: It could be interesting to see the time comparison between the proposed method and the GAT.
> > A8: Thanks for the valuable suggestion! We have added the time, memory comparison table in Appendix A.9, our experiment results show our AFGNN models need less time and memory consumption. According to our results (Table 6, 7), on Cora and Citeseer, GAT's time cost is at least three times of AFGNN's time cost, and GAT’s memory cost is two times of AFGNN's memory cost. GAT does not have recorded time and memory cost for Pubmed dataset because it requires too much memory cost and is not able to run on GPU. Therefore, we claim that AFGNN needs less time and memory cost than GAT.
> >
> > Q9: For the graph filter discriminant analysis, is it fair to compare the learned layer with the other base filter using the GFD score? Since the learned layer is picked with the highest GFD score. Maybe one or two sentences on this will be helpful.
> > A9: First, we’d like to clarify that we do not pick the best filter from the filter family. Instead, the combination weights (alpha) are learned in an end-to-end fashion on the training dataset, while the evaluated GFD scores are calculated on the test dataset. Therefore, it’s not guaranteed that a learned filter will definitely generalize better than all the base filters. Second, this experiment is to verify whether we can learn an optimal filter adaptively instead of using a fixed filter. For different datasets, there exist different optimal base filters (for example, column normalization is the best for SmallRatio and row normalize is the best for benchmark citation network), and our algorithm can indeed learn a good combination of them that generalizes well, as we expected.
> >
> > Q10: The writing of the paper must be improved. Too many typos and grammar problems will impair the presentation and the reader can be distracted.
> > A10: Thank you for pointing them out.  We have carefully proofread our paper again and polished the paper to alleviate typos and grammar errors.

---

### Official Review · AnonReviewer1 · 2019-10-28
**Official Blind Review #1**

**Rating:** 3

**Review:**

This paper introduces an assessment framework for an in-depth analysis of the effect of graph convolutional filters and proposes a novel graph neural network with adaptable filters based on the analysis. The assessment framework builts on the Fisher discriminant score of features and can also be used as an additional (regularization) term for choosing optimal filters in training. The assessment result shows that there is no single graph filter for all types of graph structures. Experiments on both synthetic and real-world benchmark datasets demonstrate that the proposed adaptive GNN can learn appropriate filters for different graph tasks.

The proposed analysis using the Fisher score is reasonable and interesting, giving us an insight into the role of graph filters. Even though the analysis is limited (using simple graph models and filter family) and the result is not surprising (given no free lunch theorem, there is very likely to be no single silver bullet fo graph filters), I appreciate the analysis and the result. But, I have some concerns as follows.

1) The proposed GNN and the optimization process
The proposed method is to extend CNN to a simple linear combination of different filter bases with learnable weights, which I don't think is very novel. Adding the GFD score as an additional constraint term is interesting, but the way of optimizing the whole objective function is unclear. (In addition, I think calling it the "regularization term" is inadequate since the term actually involves data observation, rather than a prior on parameters only.)
In the case of AFGNN_inf, I don't think it is equivalent to applying infinite lamda. If lamda is infinite, L_CE needs to be completely ignored. This needs to be clarified.
In the case of AFGNN1, I don't clearly understand how the whole objective function is properly optimized with fixed data representation. Is it also iteratively optimized? I hope this is also clarified in more detail.

2) Unconvincing experiments
The results on three real datasets do not show significant gains, and two of them are even worse than those of GAT. Furthermore, inductive learning (e.g., protein-protein interaction (PPI) dataset used in GAT) is not tested, which I think needs to be also evaluated. While two synthetic datasets (SmallGap and SmallRatio) created by the authors show significant improvement, these datasets appear to be extreme and unrealistic and look carefully selected in favor of the proposed method. I recommend the authors use for evaluation more realistic datasets that can be found in related research.


**Experience Assessment:**

I have read many papers in this area.

**Review Assessment: Checking Correctness Of Derivations And Theory:**

I assessed the sensibility of the derivations and theory.

**Review Assessment: Checking Correctness Of Experiments:**

I carefully checked the experiments.

**Review Assessment: Thoroughness In Paper Reading:**

I read the paper at least twice and used my best judgement in assessing the paper.

---

> ### Author Response · Authors · 2019-11-11
> **Response to Reviewer1**
>
> We appreciate your valuable comments and we are now actively working on the supplementary experiments on real-world datasets!

---

> > ### Author Response · Authors · 2019-11-15
> > **Response to Reviewer1**
> >
> > Thank you for your valuable feedback! We improved our paper based on your advice (we marked the modifications related to your suggestions with orange text, and highlighted the previous version with strikethrough)：
> > https://drive.google.com/file/d/1qAe72_w9Zn_mXg7rwu25o54kcQ6QhsME/view?usp=sharing
> >
> > The reviewer is majorly concerned about whether our pointed problem indeed exists on real-world datasets, and whether our proposed method can solve it. To alleviate this concern, we added a new real-world dataset and show consistent results with our analysis, which is discussed in Q1.
> >
> > Now we address your comments and concerns in detail:
> >
> > Q1: Synthetic datasets appear to be extreme and unrealistic and look carefully selected in favor of the proposed method.
> > A1: The problem we found out is not unrealistic and carefully selected, but indeed appear in real-world datasets. Each synthetic dataset we choose corresponds to a specific graph data property we analyze in Section 3.2, including "small density gap" (i.e., the graph structure is not highly correlated with labels) and "large label ratio gap" (i.e., classes are imbalanced). These properties widely occur in real-world datasets. To further justify this, we choose the "large label ratio gap" as an example and try to find a graph dataset that has this problem. We download a large scale academic graph called Open Academic Graph (OAG) [1][2][3] and choose two fields that have a large disparity in the number of papers: (1) “History of ideas”, which consists of 1041 papers; and (2) “Public history”, which consists of 150 papers. Obviously, these two classes are imbalanced so that the graph data has the "large label ratio gap” problem. We then compare our method with baselines on this OAG graph data (We open-source this dataset in the github. Detailed experiment settings and results are in Appendix A.10). According to our results (Table 8), our proposed AFGNN_inf achieves 88.22 macro F1-score, which outperforms all the other baselines (our macro F1 is at least 3% higher than all the baselines). Such a result on the real-world dataset is consistent with what we achieve on the same synthetic dataset, which indicates that the problem we reveal is not “unrealistic”, but exists in real-world datasets. The widely adopted benchmark graph datasets (cora, citeseer, pubmed), however, do not have these potential problems. Thus our analysis can also benefit the GNN research community to find other representative benchmark datasets for the node classification task.
> >
> > Q2: Model is not novel
> > A2: We’d like to emphasize that the main contribution of our work is the proposed graph filter assessment tool (GFD score) and the insights we found with the tool, which provides a unique perspective in understanding why GNN will work and how we should choose graph filters for graph data with different properties. The AFGNN model is our first attempt to learn a flexible filter that is adaptive to the graph data by leveraging the GFD score, which has successfully demonstrated the power of our assessment tool. The basic idea of the AFGNN is simple, but it works well with much less memory and time consumption than the sophisticated model as GAT. According to our results (Table 6 and 7), on Cora and Citeseer, GAT's time cost is at least three times of AFGNN's time cost, and GAT’s memory cost is two times of AFGNN's memory cost.
> >
> > Q3: "Regularization term" is inadequate.
> > A3:  Thanks for pointing it out. The standard definition of regularization is: regularization is the process of adding information in order to solve an ill-posed problem or to prevent overfitting. Our GFD is added to the loss function to guide the learning process of the filter and to avoid overfitting, therefore, previously we call this GFD term a regularization. To avoid confusion, we have changed it into “GFD loss”.
> >
> > Q4: $AFGNN_{infinity}$ is not equivalent to applying infinite lambda
> > A4: Thanks for pointing it out, we agree that our previous writing in this part is not accurate enough. We use the previous writing to emphasize we optimize $\alpha$ and $W$ iteratively by minimizing CrossEntropy loss and GFD loss respectively. We have improved our writing now.
> >
> > Q5: For $AFGNN_1$, is it also iteratively optimized as $AFGNN_{infinity}$?
> > A5: $AFGNN_1$ and $AFGNN_{infinity}$ are different. For $AFGNN_1$, we learn $\alpha$ and W simultaneously by directly minimizing the overall objective function (=CrossEntropy loss + GFD loss), but for AFGNN_inf, we learn $\alpha$ and $W$ separately. We have improved our writing for this part to make it more clear.

---

> > > ### Author Response · Authors · 2019-11-15
> > > **Response to Reviewer1 (cont.)**
> > >
> > > Q6: The results on three real datasets do not show significant gains.
> > > A6: First, our method performs the best among all the baselines that also adopt the same base filter family, and only perform worse than GAT, which has a much wider filter family space than us. However, GAT actually requires much more computation resources than us. Compared with GAT, our model needs less time and memory costs (According to our results in Table 6 and 7, GAT's time cost is at least three times of AFGNN's, and GAT’s memory cost is two times of AFGNN's.). Also, our model can deal with class imbalance issues much better than GAT (the performance of GAT on SmallRatio and Imbalanced OAG are not good, our $AFGNN_{infinity}$ achieves 93.8 and 96.3 on these two datasets, while GAT  only achieves 82.1 and 95.1). Second, currently we only use a family of simple base filters, and the performance of AFGNN is expected to be further improved by enlarging the filter family. We leave the design of the filter family as future work. Finally, we want to emphasize again that this paper’s biggest contribution is to understand and evaluate GNNs’ filter rather than to propose another GNN model. We find some hard cases that existing GNNs with fixed filter can not handle well, and propose AFGNN to enhance the performance under these hard cases. Existing benchmark real-world datasets are not sensitive enough to differentiate existing baseline models, so we also analyze how to find a more powerful dataset that can differentiate different models and generate synthetic datasets based on our analytics.
> > >
> > > Q7: Inductive learning (e.g., PPI) is not tested.
> > > A7: Thanks for pointing out. Our current proposed model is designed mainly for transductive semi-supervised node classification, and may have some limitations for the inductive setting in some cases. But our analysis result can help design the inductive filter learning model.
> > >
> > > Our analysis assumes that for a set of feature information (X), structure information (A), and their dependency relationship with labels (A|Y and X|Y), there should exist an optimal filter, and our algorithm is designed to learn a good filter for a single graph, which can approximate such optimal filter. For a transductive setting (for example, Cora, Citeseer, Pubmed), where we only need to deal with a single graph, our algorithm has shown to be effective to learn a good filter for it. For an inductive setting (for example, PPI dataset), where we are dealing with multiple graphs with different graphs, the structure information (A) is different for each graph, and thus the optimal filter for each graph can be different. Since our current algorithm can only learn a single filter for all the graphs, if the optimal filters for testing graphs are different from what the ones for training graphs, our current algorithm cannot improve the performance too much. For cases where the graph structure property doesn’t change too much, we can assume that there still exists a single optimal filter and our current algorithm can generalize well.
> > >
> > > We evaluate our model on PPI and found our AFGNN obtains better performance than all the other baselines but is worse than GAT. This is mainly because PPI have different chemical graphs that have totally different structures, and thus fall into the first case where our algorithm that only learns a single filter cannot improve the performance too much.
> > >
> > > In spite of the limitation of our current model, we’d like to point out that it is feasible to adopt our analysis result for designing an inductive filter learning algorithm. Since we’ve already found that we can infer the optimal filter based on the graph data’s properties (e.g., label imbalance will benefit column normalization, etc), we can design an model (f) that takes these graph properties as input and infer the optimal alpha, instead of learning alpha from scratch for a new graph. If we can train the f with graphs with various properties, ideally it should learn to get optimal filter for any kind of graph data. In this way, such a model can be well suitable for inductive node classification. Therefore, our analysis can still be useful to deal with inductive node classification and even other graph-related tasks. Since such model improvement is out of range to what we want to focus on in this paper, we leave it as future work.
> > >
> > >
> > >
> > > [1] https://www.openacademic.ai/oag/
> > > [2] Fanjin et al. OAG: Toward Linking Large-scale Heterogeneous Entity Graphs. In Proc. of the 25th ACM SIGKDD International Conference on Knowledge Discovery and Data Mining (KDD'19)
> > > [3] Arnab Sinha, et al. An Overview of Microsoft Academic Service (MAS) and Applications. In Proc. of the 24th International Conference on World Wide Web (WWW ’15)

---

### Public Comment · ~Yilun_Jin1 · 2019-09-30
**Interesting paper that provide direct and explainable insight towards GNN**

This paper tackles the problem of GNN property and explainability, by studying how graph convolution kernels discriminate nodes from different classes.

I think the primary novelty in this paper is that it provides easy and straightforward interpretation on GNN convolution kernels, which has been previously thought of as hard to depict. In addition, the techniques and intuitions are extremely straightforward and elegant, which is really a surprise.

---

> ### Author Response · Authors · 2019-10-02
> **Thank you!**
>
> Thank you for your comment and your interest in our work!

---

### Author Response · Authors · 2019-09-30
**Corrections of Formula Typo**

We've found a typo when we define training loss in formula 5. The GFD score should be the cumulative negation GFD score of the filter in each layer with respect to its previous layer's output. Previously we missed the GFD. The corrected version is in https://github.com/conferencesub/ICLR_2020/blob/master/DissectingGNN_ICLR%20(3)%20(1).pdf

Sorry for the mistake.

---

### Public Comment · ~Deli_Chen1 · 2019-10-10
**Two questions for this work**

Thank you for the nice work. I really appreciate the idea of GFD score and the detailed analysis. Here are some aspects I care about.

1. The improvement of AFGNN seems marginal on the real dataset.
Although I like the idea of AFGNN, which combines different filters and adaptively learn a graph-specific one, the performance improvement of it on the real dataset (CORA/CiteSeer/Pubmed) seems marginal according to Table1. (BTW, is this result statistically significant?)  While in the two manual dataset, the results are excellent. So what is the reason for the performance gap? And can AFGNN be extended to be more suitable for the actual data?

2. About the dataset split.
For the three benchmark datasets, you adopt the same setting of (Kipf & Welling, 2017). I guess it should be 20 samples each class for training and 30 samples each class for developing?  Is your dataset split fixed in all the experiment? Existing work [1] has proven that the split of dataset has a significant influence on the classification result. So I think using the mean results of multi-splitting methods may be a better idea for the node classification task likes [2][3].

[1]Oleksandr Shchur, Maximilian Mumme, Aleksandar Bojchevski, Stephan Günneman: Pitfalls of Graph Neural Network Evaluation
[2]Sun, K.; Koniusz, P.; and Wang, J Fisher-Bures Adversary Graph Convolutional Networks.
[3]Deli Chen, Yankai Lin, Wei Li, Peng Li, JieZhou, Xu Sun: Measuring and Relieving the Over-smoothing Problem for Graph Neural Networks from the Topological View

---

> ### Author Response · Authors · 2019-10-11
> **Re: Two questions for this work**
>
> Thank you so much for the comment!
>
> The key point of our work is to analyze the GNNs for node classification from a data perspective. We pointed out that there’s no best GNN filter for all datasets, and we proposed a GFD score that can assess the power of filter and help to find the optimal filter for a given dataset as well.
>
> For question 1: We proposed the AFGNN to verify our analysis, according to the experiment result, for whichever dataset, the performance of our proposed AFGNN is always among the best, which shows AFGNN is robust and effective and indicates GFD can help to select the best filter for a given dataset. Also, the current benchmark datasets cannot clearly differentiate different graph neural networks (as shown in Table 2, while the order is the same, scores of different filters are close to each other). So in our work, we identify some challenging cases for existing GNNs, then create corresponding synthetic benchmark datasets to test all GNN models. It will also guide us to look for real-world graph data to serve as the new benchmark datasets.
>
> For question 2: For the benchmark datasets split, we used 20 samples each class for training, 500 samples in total for validation, and 1000 samples in total for test. This is a standard split strategy, and most existing works, including all of our baselines (GCN, GAT, GFNN, SGC), follow this strategy. It’s true that using the mean results of multi-splitting methods may help to reduce the impact of dataset partitioning on experimental results. But in order to have fair comparisons between our model and baselines, we follow the split convention.

---

### Decision · Program_Chairs · 2019-12-19

**Decision:**

Reject

**Comment:**

The paper investigates graph convolutional filters, and proposes an adaptation of the Fisher score to assess the quality of a convolutional filter. Formally, the defined Graph Filter Discriminant Score assesses how the filter improves the Fisher score attached to a pair of classes (considering the nodes in each class, and their embedding through the filter and the graph structure, as propositional samples), taking into account the class imbalance.

An analysis is conducted on synthetic graphs to assess how the hyper-parameters (order, normalization strategy) of the filter rule the GFD score depending on the graph and class features. As could have been expected there no single killer filter.

A finite set of filters, called base filters, being defined by varying the above hyper-parameters, the search space is that of a linear combination of the base filters in each layer. Three losses are considered: with and without graph filter discriminant score, and alternatively optimizing the cross-entropy loss and the GFD; this last option is the best one in the experiments.

As noted by the reviewers and other public comments, the idea of incorporating LDA ideas into GNN is nice and elegant. The reservations of the reviewers are mostly related to the experimental validation: of course getting the best score on each dataset is not expected; but the set of considered problems is too limited and their diversity is limited too (as demonstrated by the very nice Fig. 5).

The area chair thus encourages the authors to pursue this very promising line of research and hopes to see a revised version backed up with more experimental evidence.